# Current Clinical Trials Protocols and the Global Effort for Immunization against SARS-CoV-2

**DOI:** 10.3390/vaccines8030474

**Published:** 2020-08-25

**Authors:** Gabriel N. A. Rego, Mariana P. Nucci, Arielly H. Alves, Fernando A. Oliveira, Luciana C. Marti, Leopoldo P. Nucci, Javier B. Mamani, Lionel F. Gamarra

**Affiliations:** 1Hospital Israelita Albert Einstein, São Paulo 05652-900, Brazil; gnery.biomedicina@gmail.com (G.N.A.R.); ariellydahora1997@gmail.com (A.H.A.); fernando.anselmo@einstein.br (F.A.O.); luciana.marti@einstein.br (L.C.M.); javierbm@einstein.br (J.B.M.); 2LIM44-Hospital das Clinicas HCFMUSP, Faculdade de Medicina, Universidade de São Paulo, São Paulo 01246-903, Brazil; mariana.nucci@hc.fm.usp.br; 3Centro Universitário do Planalto Central UNICEPLAC, Brasília 72445-020, Brazil; leopoldo.nucci@gmail.com

**Keywords:** COVID-19, SARS-CoV-2, immunization, vaccine, research and development (R&D)

## Abstract

Coronavirus disease 2019 (COVID-19) is the biggest health challenge of the 21st century, affecting millions of people globally. The outbreak of severe acute respiratory syndrome coronavirus 2 (SARS-CoV-2) has ignited an unprecedented effort from the scientific community in the development of new vaccines on different platforms due to the absence of a broad and effective treatment for COVID-19 or prevention strategy for SARS-CoV-2 dissemination. Based on 50 current studies selected from the main clinical trial databases, this systematic review summarizes the global race for vaccine development against COVID-19. For each study, the main intervention characteristics, the design used, and the local or global center partnerships created are highlighted. Most vaccine developments have taken place in Asia, using a viral vector method. Two purified inactivated SARS-CoV-2 vaccine candidates, an mRNA-based vaccine mRNA1273, and the chimpanzee adenoviral vaccine ChAdOx1 are currently in phase III clinical trials in the respective countries Brazil, the United Arab Emirates, the USA, and the United Kingdom. These vaccines are being developed based on a quickly formed network of collaboration.

## 1. Introduction

In December 2019, in the Chinese city of Wuhan in China’s Hubei province an outbreak related to severe acute respiratory syndrome coronavirus 2 (SARS-CoV-2)—that is, a virus genetically close to bat-CoV-RaTG13 and bat-SL-CoVZC45—emerged, leading to coronavirus disease 2019 (COVID-19). This infectious disease caused an unprecedented health emergency, which was declared a pandemic by the World Health Organization (WHO) on 11 March 2020 [1]. 

Subsequently, new approaches to vaccine development or the adoptive transfer of immunity were rapidly implemented in preclinical and clinical studies. These studies aimed to avoid infection and prevent the symptoms of COVID-19, a disease of heterogeneous symptoms caused by SARS-CoV-2. As of the end of June 2020, this virus had affected about 10 million people globally and killed about 560,000 individuals [2].

Currently, there is not yet a consolidated protocol or therapeutic strategy approved for COVID-19 critical patients [3]. At this moment, the best practices for the supportive management of respiratory acute hypoxemic failure must be used. 

The initial treatment of COVID-19 was restricted to overall supportive care and critical care because no other appropriate therapies or vaccines existed. However, many clinical trials are investigating the effect of anti-inflammatories, antimalarials, antivirals, plasma therapy, cardiovascular drugs, antibiotics, cell therapy, and cancer drugs among other medications for COVID-19 treatment [4,5]. Dexamethasone and anticoagulant are suggested since some severe, critical, and deceased COVID-19 patients have significant coagulation dysfunction with increased D-dimer concentration, decreased platelet counts, and the prolongation of prothrombin time [6]. Regarding the prophylactic use of Bacilo Calmette–Guérin (BCG) (Pasteur Institute at Lille, France) vaccine against SARS-CoV-2, it was observed that previous BCG (Lille, France) immunization correlates with a lower incidence and gravity of the COVID-19 disease across different countries, even when the BCG (Lille, France) immunization was performed in childhood. Antimalarials such as chloroquine and hydroxychloroquine are being extensively investigated, but no strong evidence of their benefits has been released yet. Thus, there are no specific mechanisms or robust evidence of the efficacy of these therapeutic procedures [7]. 

Thus, during the pandemic caused by COVID-19, several vaccine candidates with attenuated virus, encoding, or presenting SARS-CoV-2 antigens have been developed globally, reaching clinical trial phases I or II for the evaluation of their safety and immunogenicity. These vaccines include those based on inactivated virus [8,9,10,11,12]; on nucleic acid platforms as messenger ribonucleic acid (mRNA) [13,14,15,16,17,18,19,20], self-amplifying RNA (saRNA) [21], and DNA/plasmids [22,23,24,25,26,27]; recombinant adenovirus serotypes platforms as adenoviral vector 5 [28,29,30,31,32], chimpanzee adenoviral vector ChAdOx1 [33,34], and combined serotypes vectors 5 and 26 [35,36]; recombinant viral protein subunits [37,38,39,40,41,42,43,44]; modified dendritic cells [45,46,47]; artificial antigen-presenting cells [48]; and virus-like particle (VLP) vaccine [49]. Six protocols are developing phase II and/or III clinical trials using the chimpanzee adenoviral vector ChAdOx1 [50,51,52], purified inactivated SARS-CoV-2 vaccine [53,54], and mRNA-1273 vaccine [55]. Several of these vaccines undergoing trials comprise new technology that has not been tested previously.

The advent of nanotechnology enables different mechanisms to target viruses—for instance, through the use of acid functionalized multi-walled carbon nanotubes comprising photo-activated molecules [56], or natural (such as chitosan) or synthetic (such as polyethyleneimine (PEI)) polymeric nanoparticles [57]. These kinds of nanomaterials are able to work as gene carriers and have the potential to deliver small interfering RNA (siRNA) in COVID-19 patients [57]. In addition, nanotechnology has enabled the development of new vaccines, including vaccines based on recombinant protein nanoparticles with or without Matrix M^TM^ adjuvant [42], and mRNA vaccines through the encapsulation of modified nucleoside mRNA inside lipidic nanocapsules [17,18,19,20,58], which confers plasticity in terms of antigen manipulation and a potentially rapid effect. Intense efforts are underway to expand the technological knowledge against COVID-19 as part of a global scientific effort to battle this virus. However, to date no review has been conducted about this global development effort and the associated cooperation networks used to test new vaccines. This lack of information empowers the movements of anti-vaccine groups [59].

Nanotechnological mRNA-1273 arises in clinical trials just two months following SARS-CoV-2 sequence identification and is currently one of the most advanced vaccines already in phase III clinical trial protocols (CTPs) [55]. Viral vector platforms enable protein overexpression, and their long-term stability can induce robust immune responses with a single dose. At present, the chimpanzee adenoviral vector from Oxford (ChAdOx1) is another advanced vaccine in study phase (II-III) [51,52] and phase III [50]. In addition, recombinant protein subunits, as immunologic adjuvants that use a squalene (TLR4) agonist [60], previously licensed to be delivered as vaccines for other conditions, can provide apparatus for the necessarily large-scale production of the newly developed vaccines [61].

The majority of the recently developed vaccines aim to stimulate antibody synthesis with the necessary potential to neutralize the SARS-CoV-2 spike protein (S protein) to prevent the uptake of the single-strand RNA of this virus in ACE2 receptor positive cells [62].The exceptions are the vaccines using an attenuated/inactivated whole virus, which would lead to broader immunization [10,11,63]. In addition, vaccines using antigen presenting cells and artificial antigen presenting cells (aAPC) will elicit a robust cellular immune response in addition to humoral immunity.

This review aims to highlight the global scientific effort in the fight against SARS-CoV-2 for the development of new immunization approaches through either conventional or novel vaccine strategies against SARS-CoV-2.

## 2. Methods

### 2.1. Search Strategy

This systematic review followed the Preferred Reporting Items for Systematic Reviews and Meta-Analyses (PRISMA) Guideline [64]. We conducted a search for published protocols until 26 July 2020 in the subsequent clinical trial databases: ClinicalTrials.gov, the Chinese Clinical Trial Registry (ChiCTR), the European Union Clinical Trials Register, and the WHO COVID-19 global research database [65]. Next, we applied the keyword sequence (COVID-19 OR SARS-CoV-2) AND (vaccine OR immunization) in the search fields of these databases.

### 2.2. Inclusion and Exclusion Criteria

This review included only clinical trial protocols that addressed the development of vaccines for preventing infections caused by the SARS-CoV-2 virus.

The reasons for excluding studies were as follows: (i) clinical trial protocols for observational study; (ii) clinical trial protocols that were canceled or not approved until the date of searching in the analyzed databases; (iii) clinical trial protocols that used drugs for COVID-19 treatment not associated with a vaccine; (iv) clinical trial protocols that use existing vaccines for other diseases to prevent COVID-19—for example, BCG; and (v) clinical trials using plasma for therapeutic protocols.

### 2.3. Data Extraction, Data Collection, and Risk of Bias Assessment

In this systematic review, six authors (G.N.A.R., A.H.A., M.P.N., L.P.N., J.B.M., and F.A.O.) organized in pairs independently and randomly reviewed and evaluated the information recorded from the clinical trial protocols identified by the search strategy in the databases mentioned above. These same authors evaluated the protocols to decide whether the eligibility criteria were met. The discrepancies in the study selection and data extraction between the six authors were discussed with two other authors (L.C.M. and L.F.G.) and resolved.

G.N.A.R., M.P.N., F.A.O., L.P.N., A.H.A., and J.B.M. analyzed studies of active immunization for COVID-19 through vaccines. After the study selection, the authors G.N.A.R., L.C.M, and M.P.N. analyzed the description of the experimental vaccine; M.P.N., F.A.O., and J.B.M. analyzed the study design, research arms, and applied interventions; and A.H.A and L.P.N analyzed the collaborative network between companies and universities as well as the partnerships established between different centers and hospitals. The analysis of the clinical trials and the preparation of the tables were carried out by consensus. For each case of disagreement, two senior authors (L.C.M. and L.F.G.) decided on the addition or subtraction of data. The final inclusion of studies in this review was agreed with all the authors.

### 2.4. Data Analysis

All the results were described and presented using the percentage distribution for all the variables analyzed in the tables.

## 3. Results

### 3.1. Study Selection

After applying the search strategies in the databases, 200 clinical study protocols were identified (171 protocols in ClinicalTrials.gov, 13 in ChiCTR, 12 in the EU Clinical Trials Register, and 4 in the WHO COVID-19 global research database). The search strategy used the Preferred Items for Reporting Guideline for Systematic Reviews and Meta-Analyses (PRISMA) [64]. Based on the established inclusion and exclusion criteria, of 171 protocols identified in ClinicalTrials.gov, 135 clinical trials were excluded after screening (68 BCG vaccine protocols, 49 observational protocols, 1 suspended, and 17 that used convalescent plasma for passive immunization), leaving 36 protocols selected from these databases. Of the 13 protocols identified in ChiCTR, 5 were excluded after the analysis (3 were observational protocols, 1 was a canceled/not approved clinical trial protocol, and 1 protocol was described as an immune response induction method not specific for COVID-19/SARS-CoV-2), leaving 8 selected clinical trial protocols. Finally, of the 12 protocols registered in the EU Clinical Trials Register, 11 were excluded (7 BCG vaccine protocols, 2 using convalescent plasma, and 2 protocols that were also registered in ClinicalTrials.gov), leaving only 1 study selected from this database. In total, 50 clinical trial protocols for active immunization for COVID-19 through vaccine application [8,9,10,11,12,13,14,15,16,17,18,19,20,21,22,23,24,25,26,27,28,29,30,31,32,33,34,35,36,37,38,39,40,41,42,43,44,45,46,47,48,49,50,51,52,53,54,55,66,67] were included in the present work.

### 3.2. Overview of Clinical Trial Protocols for Active Immunization for Covid-19

The selected clinical trials focused on interventional approaches for active immunization through the application of the various types of vaccines currently being developed against SARS-CoV-2 (Table 1). 

The distribution of these CTPs over the time by phase (Figure 1) shows that the phase I studies started in February. The number of new studies by month (Figure 1A) shows that the number of CTPs in phase I was triplicate and quadruplicate in March and April, respectively, as did the beginning of phase II and phase II-III concomitant studies. In May, the number of studies at phase I reduced to a single CTP, and the I-II phase studies declined by half compared to the previous month. In addition, a new study started phase III. In June, the total number of new CTPs was the same as presented in April, but half of the studies were phase I and the other half were phase I-II. In July, as well as in June, there were six new studies at phase I-II and a study in phase II, but the number of CTPs at phase I decreased by 33.3%. There is a clear oscillation in the number of CTPs at phase I over the months, less marked in CTP at phase I-II. At the same time, there occurred an increased number of new studies in phase III.

A cumulative analysis of the number of CTPs by phase (Figure 1B) shows a continuous increase in phase I-II CTPs, which also occurred in phase I studies with a slight oscillation between April and May. On the other hand, cumulatively, the number of phase II, II-III, and III studies has remained constant since March, with a slight increase in phase III CTPs between June and July.

### 3.3. Properties and Features of Vaccines against COVID-19

#### 3.3.1. Nucleic Acid Vaccines

Regarding the CTPs targeting vaccine development and application, their platforms display a variety of characteristics and properties (Figure 2; Table 1). Nucleic acid vaccines are the ones most tested in CTPs at this moment (32%). They are based on a genetically engineered plasmid containing DNA (12% of the CTPs) or RNA (as mRNA and saRNA, comprising 20% of the CTPs) (Figure 2) usually encoding a selected antigen (mainly S protein). 

##### RNA Vaccines

The mRNA was used in 18% of the selected clinical trials in different phases (I, I-II, II), including a phase III mRNA1273 vaccine CTP [55]. The mRNA1273 [17,20,55] and BioNTech 162 (BNT162) [14,15,18,19] vaccines used a lipid nanoparticle (LNP) encapsulated nucleoside modified mRNA encoding the S protein of SARS-CoV-2. 

BNT162 by the BioNTech (BNT) and Pfizer companies embrace different RNA vaccine candidates, comprising two nucleoside modified mRNA (modRNA) vaccines, an uridine containing mRNA (uRNA) vaccine, and a saRNA vaccine. Besides this, two of them codify full-length S protein, and the others codify S protein receptor binding domain (RBD) [68]. Initially, studies with BNT were developed with the four mentioned BNT162 vaccines; however, more recently a CTP with China as the recruiting country applied only the BNT162b1 vaccine [14]. Other CTPs of the same vaccine are recruiting patients in Germany [19] and the USA [18]. BNT162 is being administered in a prime/boost regimen [14,15,18,19], in which the same immunogen is applied in the prime, and the booster CTP objective is to observe the immunogenicity of the subjects against S protein and RBD through IgG titration in healthy individuals [14,15,18,19]. 

The CTPs using mRNA1273 by Moderna TX Inc are all being completely developed in the USA [17,20,55]. This vaccine encodes full-length S protein. At phase I, this study tested different concentrations of mRNA at 10, 25, 50, 100, and 250 µg in different interventional arms [17]. In the phase III intervention, the established dose was 100 µg via intramuscular administration in two doses 28 days apart in healthy individuals over 18 years old [55]. 

The phase I German CureVac´s novel coronavirus (CVnCoV) CTP [16] and another mRNA vaccine CTP recruiting in China [13] encode an unspecific protein and an RBD S protein of SARS-CoV-2, respectively, but there is no information about the delivery systems. Recently, an vaccine based on self-amplifying RNA encapsulated by lipid nanoparticle (LNP-nCoVsaRNA) phase I used saRNA encoding S protein in the United Kingdom, which will produce more antigen per transfected cell due to the replicative element [21]. 

##### DNA Vaccines

DNA vaccines [22,23,24,25,26,27] were used in 12% of CTPs with different delivery systems. Plasmids encoding S protein (pGX9501) are delivered with CELLECTRA@2000 electroporation system in a phase I trial in the USA, with a 0.5 mg/dose and 1 mg/dose [27], and in a phase I-II trial in the Republic of Korea, with a 1 mg/dose and 2 mg/dose delivered in a population ranging from 18 to 64 years old in sequential assignment [24] in two doses, as shown in Table 2. 

The Canadian bacTRL-Spike-1 vaccine uses a genetically modified probiotic *Bifidobacterium longum* as a delivery system for a synthetic DNA plasmid encoding S protein to host immune cells in a phase I study; this is the first time that bacTRL has been delivered in humans orally with varied doses in parallel assignment containing 1, 3, or 10 billion colony-forming units [26].

The GX-19 vaccine plasmid in a phase I-II trial in the Republic of Korea by Genexine, Inc. Company (Gyeonggi, Korea) [25] is applied via intramuscular administration in a dose-escalation protocol 28 days apart. A second COVID-19 Indian vaccine ZYCOV-D by Cadila Healthcare Ltd. (also known as Zydus Cadila, Ahmedabad, India) is testing the vaccine in two groups with different age ranges, one from 12 to 55 and another from 12 to 65 years old. In both groups, the doses of 0.5 mL are delivered intramuscularly 14 days apart [22]. The CTP for the Japanese AG0301 DNA vaccine delivers the doses in the same period of time as ZYCOV-D [23], with a population ranging from 20 to 65 years old. 

#### 3.3.2. Non-Replicating Viral Vector Vaccines

The second most used platform comprises different serotypes of non-replicating recombinant adenoviral vectors (24%) that comprise the chimpanzee adenoviral vector ChAdOx1 nCoV-19 by Oxford University (Figure 2) [33,34,50,51,52], currently with two in phase II-III [51,52] and another in phase III [50] in the United Kingdom and Brazil, respectively. The adenovirus type 5 (Ad5-nCov) by CanSino Biologics [28,29,30,31,32] targeting the full-length S protein of SARS-CoV-2 displays two studies in phase II in China [29,32].

The Russian Gam-COVID-Vac by Gamaleya Research Institute, which combines Ad5 and a recombinant adenovirus type 26 (Ad26) vectored system targeting the S protein of SARS-CoV-2 [35,36], is in phase I-II. In one of these CTPs, the Gam-COVID-Vac solution is lyophilized [36]. The most advanced ChAdOx1 nCoV-19 protocol is delivered in a single dose of 5 × 10^10^ viral particle [50,51], which is being tested two times on days 1 and 28 [51]. A clinical trial divided the groups accordingly to the age of the participants: children ranging from 5 to 12, adults from 18 to 55, and people older than 56 years of age [52]. In turn, the age used in two most advanced Adn5-nCoV CTPs is older than 18, with single doses of 5 × 10^10^ and 1 × 10^11^ in a study with crossover assignment [32] and parallel assignment [29], both with three interventional arms, as shown in Table 2. 

#### 3.3.3. Inactivated Virus Vaccines

The inactivated SARS-CoV-2 vaccine was used in 18% of the selected clinical trials (Figure 2) [8,9,10,11,53,54,66,67]. The vaccine developed by SinoVac Biotech Corporation is classified in the CTPs as purified inactivated SARS-CoV-2 and is an adsorbed vaccine. According to preclinical study information, this vaccine, named PiCoVacc, was developed by propagating the CN2 strain of SARS-COV-2 inside VeroCells and inactivated using the chemical agent β-propiolactone [69]. This Chinese vaccine is being tested in Brazil in a phase III study [54] and in a phase III CTP developed in the United Arab Emirates [53]. Another purified inactivated vaccine also propagated SARS-CoV-2 inside VeroCells [8,9,53], but there is no information about the strain or inactivating agent used in pre-clinical studies. A recent phase I-II Indian vaccine BBV152 will apply three inactivated whole-virion strains designated BBV152A, BBV152B, and BBV152C [66]. Another two studies developing purified inactivated SARS-CoV-2 vaccine provides no information about the inactivation processes [12,67]. All these vaccines target multiple proteins of SARS-CoV-2. The most advanced inactivated vaccine (phase III) is applied in two doses 14 days apart in the deltoid muscle [54] in participants of 18 years onwards, as shown in Table 2. 

#### 3.3.4. Protein Subunit Vaccines 

Vaccines based on the recombinant protein subunit are being tested in 16% of the selected CTPs (Figure 2) [37,38,39,40,41,42,43,44], identified by different names according to their features. The most advanced of them (phase II) apply SARS-CoV-2 protein subunits engineered, produced, and secreted by Chinese Hamster Ovary (CHO) cells [39,41]. Phase I SCB-2019 vaccine is a recombinant SARS-CoV-2 trimeric S protein subunit vaccine, and is tested with or without lipid-based Adjuvant System 03 (AS03) that increases the reaction of the innate immune system, or with a oligonucleotide sequence containing CpG motif (CpG 1018) adjuvants plus aluminum, with a CTP being developed in Australia [43]. Another Australian phase I vaccine is NVX-CoV2373, a recombinant S protein nanoparticle vaccine tested with or without the Novavax saponin-based Matrix M^TM^ adjuvant [42], COVAX-19^TM^, a Vaxine proprietary Advax™ adjuvant technology combined with a recombinant S protein [38,44] and a S protein stabilized using the molecular clamp technology applied with MF59 immunologic adjuvant [37]. A CTP is being developed based on the production of the RBD of S protein through fast-growing tobacco plant technology [40]. The use of adjuvants increases the activity of T and B cells mediated by antigen-presenting cells (APCs) [70]. 

#### 3.3.5. Dendritic Cells Vaccines

Another 6% of clinical trials (phase I-II) are testing vaccines based on dendritic cells (DC) (Figure 2) [45,46,47]. One of them, from the USA, is testing subcutaneously autologous DC loaded with 1× or 3.33% antigens from SARS-CoV-2 with or without 250 μg or 500 μg of granulocyte-macrophage colony-stimulating factor (GM-CSF), targeting non-specific proteins. This vaccine is developed with the isolation of monocytes from heparinized blood incubated with IL-4 and GM-CSF that are differentiated in vitro in DC, which in turn are incubated with SARS-CoV-2 antigens [46]. 

China is testing subcutaneously a recombinant chimeric DC vaccine targeting an unspecified SARS-CoV-2 epitope. This vaccine, called LV-SMENP DC, is applied subcutaneously (5 × 10^6^) with intravenous antigen-specific cytotoxic T cells (10^8^). The subjects will be analyzed once a week for four weeks. LV-SMENP DC was developed with a technology that modifies these cells through the lentiviral vector system NHP/TYF [45]. 

The same lentiviral vector system NHP/TYF was able to modify artificial antigen presentation cells (aAPC) that represent 2% of CTPs [48]. All these lentiviral modified cells encode multiple proteins of SARS-CoV-2, and both CTPs are being developed in China [45,48]. 

#### 3.3.6. Virus Like Particles Vaccines

Besides this, a phase I study using VLP (2% of the selected CTPs) is recruiting patients in Canada (Figure 2) [49]. This vaccine is administered 21 days apart via intramuscular injection. Each dose is applied to the deltoid region of alternated arms. The doses being tested are initially 3.75 μg in a small number of the population (healthy adults 18 to 55 years of age) with dose-escalation to 7.5 μg and 15 μg unadjuvanted or adjuvanted with either CpG 1018 or AS03, with a follow-up period of six months. After this time, the immunogenicity of the subjects will be tested.

### 3.4. Trial Progression Rate

The vaccine development clinical trials are displayed by phase, country, and trial progression rate (TPR) according to the beginning and completion date of each study (Table 1) in Figure 3. Most of the clinical trials are in simultaneous phase I-II (46%) [8,9,10,11,12,14,15,19,22,23,24,25,30,33,34,35,36,40,45,46,47,66,67]. In China the TPR ranges from 7.1% to 100% [8,9,10,11,12,14,45,47,67], in Germany the TPR of one study is 94.0% [19], in Russia it is 66.1% [35,36], in the United Kingdom the TPR is 25.25% [34], in Canada the TPR is 18.8% [30], in the USA it is 10.3% [46], in South Africa it is 10.0% [33], in the Republic of Korea the TPR ranges from 3.7% to 5.3% [24,25], in India it ranges from 3.3% to 3.7% [22,66], and in Japan it is 3.3% [23]. Then, of the clinical trials in phase I (34%) [13,16,17,18,21,26,27,28,31,37,38,39,42,43,44,48,49], in China the TPR ranges from 4.8% to 52.0% [13,16,21,26,37,38,44], in the USA it is 45.5% [28], in Canada the TPR ranges from 21.4% to 25.1% [17,27], in Australia the TPR ranges from 5.4% to 14.4% [42,48,49], in Germany it is 13.1% [31], and in the United Kingdom it is 8.8% [18].

Four clinical trials (8%) are in phase II [20,29,32,41]. In China, the trial has a TPR of 35.7%; in the USA, the trial has a TPR of 13.5%; two clinical trials (4%) are in simultaneous phase II-III [51,52] in the United Kingdom, and the studies have a TPR ranging from 18.8% to 30.1%. Four clinical trials (8%) are in phase III [50,53,54,55]. In Brazil, the TPR ranges from 5.5% to 18.9% [50,54]; in the United Arab Emirates, the TPR is 2.7% [53]; and in the USA, the trial has not yet been initiated [55]. Among all clinical trials involving vaccine development, so far only one study (2%) has been completed [10], 56% are recruiting patients [9,11,12,14,16,17,18,19,20,21,22,23,25,27,28,35,36,37,38,39,42,43,45,48,49,50,53,63,66], 30% are not yet recruiting [8,10,13,24,26,30,33,40,41,44,46,47,51,54,55], 6% are active but not recruiting [31,32,34], 4% have not reported their recruitment status [15,52], and 2% have classified their recruitment status as enrolling by invitation [67], as shown in Table 1.

### 3.5. Study Design of Clinical Trial Protocols

Regarding the clinical trial protocol studies design of vaccination against COVID-19 (Figure 4 and Table 2), 48% of the protocols were multicenter (MC) research trials and 48% were single center (SC); only two CTPs did not report this information. The estimated enrollment of clinical trials in phase III (3%) ranged from 30,000 to 2000 individuals [50]; in phase II-III (6%), there were 10,260 individuals [51,52]; in phase II (9%), it ranged from 500 to 2000 individuals [20,29,32,41]; in phase I-II (47%), it ranged from 30 to 2128 individuals [8,9,10,11,12,14,15,19,22,23,24,25,30,33,34,35,36,40,45,46,47,66,67]; and in phase I (35%), it ranged from 32 to 7600 individuals [13,16,17,18,21,26,27,28,31,37,38,39,42,43,44,48,49,50]. The number of volunteers estimated in each protocol was represented by the scale bar in Figure 4. The CTP intervention design was mostly randomized (76%) and used some type of masking (69%), which varied from 12% single blinding (participant) to 22% double blinding (participant, investigator), 16% triple blinding (participant, investigator, outcomes assessor), and 18% quadruple blinding (participant, care provider, investigator, outcomes assessor). However, certain CTPs have not adopted any techniques for minimizing the bias in allocations and blinding—20% were non-randomized; 4% did not report the strategy of allocation; 28% did not use masking (none), keeping the research open label; 4% did not report the strategy of masking. The intervention model was 60% in parallel-assignment design, following by 32% with sequential assignment design, 4% with single group assignment, 2% with crossover assignment, and 2% did not report the intervention model (Figure 4). The number of intervention arms used in the clinical trial protocols were mainly under 4 (63%), but 22% used between 5 and 10 arms; 6.1% used between 10 and 20; and 8.2% used more than 20 arms of intervention, corresponding to the type of vaccine analyzed according to the different dose concentrations (low, medium, high), the number of doses (single or more than one), and the days of administration (single or double). In addition, the arms were distributed for different ages, including less than 18 years old (8%), more than 18 years old (12%), between 18 and 70 years old (63%), and more than 70 years old (17%).

### 3.6. Global Research Network in Clinical Trial Protocols

The distribution of vaccine CTP networks between the sponsor and the collaborating institutions is highlighted in the Figure 5 map, focusing mainly on the CTPs collaborating with more than five centers. Among the vaccines leading multicenter CTPs, there is one in the USA controlled by ModernaTX Inc. at phase III (green circle) coordinating 87 collaborating institutions (green cylinders around the red bar of Figure 5), with an estimated enrollment of 30,000 participants, not yet recruited [55] (as detailed in Table 1 and Table 2). In addition, there is another CTP in the USA at phase II (blue circle) recruiting 600 volunteers from 10 centers (blue cylinders of Figure 5) [20]. Two CTPs in Europe are controlled by Oxford University in partnership with AstraZeneca [51,52], coordinating more than 20 collaborating institutions (hospitals, labs, etc., represented by the yellow and purple cylinders in Figure 5) involving approximately 10,260 participants each, at phase II or III (green-blue circle). There is another CTP at phase I-II (red-blue circle) that is recruiting 1090 volunteers at six centers in the United Kingdom [34] (gray cylinders in Figure 5). The CTP in South America is controlled by Butantan Institute in partnership with the Sinovac Life Sciences Co. (Beijing, China) [54], and is at phase III (green circle), coordinating 12 centers in Brazil (dark green cylinders in Figure 5) and recruiting 8870 participants. In Asia, the CTP controlled by Bharat Biotech International Ltd. Is at phase I-II of vaccine development, recruiting 1125 participants in 11 centers in India [66]. In Africa, the vaccine center coordinated by the University of Witwatersrand phase I-II (red-blue circle) is recruiting 2000 volunteers in six centers in South Africa [33]. All these centers leading vaccine multicenter CTPs are highlighted with enlarged figures around the map in Figure 5.

However, other centers for vaccines against COVID-19 in development in Europe, Asia, Africa, the Americas, and Oceania are highlighted in the map of Figure 5.

The global distribution of CTPs, according to the execution phase, as shown in Figure 6 and Table A1 at phase I, comprises 17 of 50 (34%) CTPs [13,16,17,18,21,26,27,28,31,37,38,39,42,43,44,48,49]. Among these 17 CTPs, 5 (27.8%) are from China [13,28,31,39,48], 4 (27.8%) are from the USA [17,27,42,44], 2 (11.1%) are from Canada [26,49], 3 (16.7%) are from Australia [37,38,43], 1 (11.1%) is from Germany [16], 1 (5.6%) is from the United Kingdom [21], and one of the CTPs is shared by Germany and the USA [18], as shown in Figure 6A. At phase I-II, there are 23 of 50 (46%) CTPs [8,9,10,11,12,14,15,19,22,23,24,25,30,33,34,35,36,40,45,46,47,66,67]. Among these 23 CTPs, most are in Asia: 9 (41.7%) are from China [8,9,10,11,12,14,45,47,67], 2 are from the Republic of Korea (8.3%) [24,25], 2 are from India (8.3%) [22,66], and 1 is from Japan (4.2%) [23]. Then, in Europe, two are in Eastern Europe from Russia (8.3%) [35,36], two are from Germany (8.3%) [19], and one is from the United Kingdom (4.3%) [34]. In North America, there are three CTPs from the USA (12.5%) [15,30,46]. There is also one from South Africa (4.3%) [33], as shown in Figure 6B. The CTPs in phase II display 4 of 50 (8%) [20,29,32,41], and they are from China (75%) [29,32,41] and the USA (25%) [20], as shown in Figure 6C. There are two phase II-III CTPs out of 50 (4%) [51,52], both located in the United Kingdom (Figure 6D). 

However, there are already 4 of 50 (8%) CTPs at phase III [50,53,54,55]. Two of them are in Brazil—one is testing the vaccine developed by the University of Oxford [50] at Universidade Federal de São Paulo (UNIFESP), and other is testing the vaccine developed by Sinovac Life Science Co. at Butantan Institute [54]; one is in the United Arab Emirates, and is testing the vaccine developed by China National Biotec Group Co. at Beijing [53]; and one in the USA, testing the vaccine developed by ModernaTX, Inc. [55], as shown in Figure 6E. 

Only 6 of 50 CTPs (12%) comprise a network with other institutions located in different countries. These include the United Kingdom and Brazil [50], China and the United Arab Emirates [53], Canada and China [30], the USA and the Republic of Korea [24], the USA and Australia [42], and Germany and the USA [18]. Four studies do not reveal about their partnerships [15,40,41,44]. The most CTPs (82%) have entered into partnerships with other institutions within the same country.

## 4. Discussion

Approximately four months after a pandemic was declared by the WHO and more than 640,000 deaths have occurred, a consolidated therapeutic drug strategy protocol approved for COVID-19 still does not exist [71]. At this moment, the epicenter of the outbreak is the American continent, mainly USA and Brazil, and isolation or social distancing remain the central WHO recommendations to avoid infection and increasing numbers of deaths [72].

Unprecedented global initiatives and new partnerships are being established to accelerate the research and development of tests and therapies to control the spread of this pandemic around the world. However, vaccines that mitigate the damage caused by this infectious disease can take much longer to be available with proven safety and efficacy [73]. The time frame for vaccine supply remains uncertain, but to date global labs and industries have registered 50 vaccine CTPs in leading research databases, using eight platforms based on inactivated virus [8,9,10,11,53,54,66,67]; nucleic acid such as mRNA [13,14,15,16,17,18,19,20,55], saRNA [21], and DNA/plasmid [22,23,24,25,26,27]; recombinant adenovirus serotypes platforms, such as adenoviral vector 5 [28,29,30,31,32], chimpanzee adenoviral vector ChAdOx1 [33,34,50,51,52], and combined serotypes vectors Adenovirus 5 and 26 [35,36]; recombinant viral protein subunits [37,38,39,40,41,42,43,44]; modified dendritic cells [45,46,47]; artificial antigen-presenting cells [48]; and VLP vaccine [49]. The advantages and limitations of these platforms will be detailed ahead.

The current landscape of COVID-19 vaccine development shows that most CTPs are being developed in Asia (Table A1). However, regarding former CTPs, only 1 of 4 (15%; [53]) phase III CTPs are located in Asia. There is one CTP recruiting in Abu Dhabi/United Arab Emirates, there is a protocol that was developed in China with an inactivated virus platform, and this protocol was also used in another phase III CTP in Brazil [54]. The virus vector platform vaccine developed in the United Kingdom (Oxford University) is being tested in the most advanced studies (phase II-III), both in the United Kingdom [51,52] and Brazil [50] (phase III). These CTPs (8% of total CTPs) have used the same chimpanzee adenoviral vectored vaccine targeting the S protein of SARS-CoV-2 [50,51,52], in which the recent results show safety and a strong immune response [74,75], but these CTPs differ due to the number of interventional arms testing different vaccine doses (number and concentration), the group age of volunteers, the method of dosing analysis (Abs260, Abs260 corrected for PS80 and qPCR), and other comparisons. Lastly, the mRNA vaccine from Moderna TX, Inc., (Cambridge, MA, USA) is a USA phase III CTP that is the biggest trial, with 30,000 subjects in 87 recruiting centers (Table A1), but the recruiting is just starting [55]. This vaccine platform used advanced technology to improve the drug delivery.

The gold standard for the success of a vaccine is related to its broad and sustained immunogenicity, with adequate safety and efficacy. For this achievement, the antigen delivery systems and their eligibility are important factors [62,76].

Due to the outbreak of COVID-19, it was essential to optimize and reduce the normal stages of the vaccine development process, but possible adverse effects may occur, especially for those belonging to high-risk groups, and this is necessary to take into account during the inclusion period of subjects in these trials [73]. This comprising steps phenomenon is expressed by comprising stages (I-II) [8,9,10,11,12,14,15,19,22,23,24,25,30,33,34,35,36,40,45,46,47,66,67] and (II-III) [51,52] and shortcuts in 50% of the current vaccine CTPs. However, in April, only 5 [17,27,31,45,48] of 50 CTPs were considered the most advanced candidate vaccines to initiate clinical development for (phase I) [73], and none of these CTPs displayed clustered phases in their trials.

On 03 April 2020, Cohen [77] identified eight different vaccine groups or “platforms” in preclinical and clinical development, classified as inactivated or attenuated whole viruses, genetically engineered proteins, and mRNA technology. At that time, the most advanced studies were in phase I. After four months, there are the same eight different vaccine platforms and CTPs (88.3%), which include inactivated virus [8,9,10,11,12], non-replicating viral vectors [28,29,30,31,32], nucleic acids (RNA [13,15,16,17,18,19,20] or DNA [25,26,27]), protein subunits [42,43,44], but now most of these CTPs are in phase II-III or III.

In this review, the non-replicating viral vector was the main technology/platform used (24%). The advantage of the vaccine based on viral vectors is the combined stimulation of the innate and adaptive immune response to the heterologous viral infection and against the antigen expressed by the vector, usually the S protein [78]. The problem related to heterologous response is the previous exposure to some adenovirus serotypes by the human population, leading to a pre-existing anti-vector response, which makes the vector a disadvantage. The use of Ad5 in immunological practice is well established and highly efficient, and exhibits simple manipulation and ease of purification. However, the specific response of the transgene can be mitigated by pre-existing adaptive immune responses to antigenic targets in the vector itself [79]. The previous results of the CTP developed by CanSino confirmed the activation of CD4 and CD8 T cells in the vaccine recipients. However, the vaccine-induced specific antibody or T cell response were partially diminished by the presence of pre-existing anti-Ad5 immunity [80]. On the contrary, the study by Zhu [80] in phase I and the study by Folegatti [75] used an Ad5 vectored COVID-19 vaccine that was very immunogenic, able to induce humoral and cellular responses in most trial participants, and the detectable immune responses was very rapid, with the T cell responses peaking at day 14 after vaccination and the antibodies peaking at day 28.

Therefore, less prevalent adenoviral serotypes, such as Ad35 and Ad26, or primate derivatives, such as chimpanzees, are more often used in vaccine development, as they mimic a natural infection at low risk of herologous response, and stimulate a significant immune response without additional adjuvants [81]. Previous studies using a replication-deficient adenovirus from chimpanzee (ChAdOx1) led to the immunization of BALB/C mice, CD1 mice [82], and rhesus monkeys [83] against MERS-CoV. This adenovirus construct, now encoding the full-length S protein of SARS-COV-2, was able to generate high titers of neutralizing antibodies and a robust CD8 T cell response against this viral protein [82]. 

DNA or RNA vaccine platforms, also known as synthetic vaccines, were identified in 32% [13,14,15,16,17,18,19,20,55] of the selected CTPs. Vaccines with antigen-encoding DNA/RNA as a platform are relatively versatile, easy to manipulate, and relatively inexpensive to synthesize compared to other types of vaccines. These vaccines have similar strategies of action—for instance, they have coding for the synthesis of target antigens that will be expressed in immune cells. However, the target of each platform is different, in addition to the advantages or risks they present [84].

mRNA based-vaccines provide benefits compared to protein subunits, inactivated or attenuated viruses, and DNA-based vaccines. These benefits were discussed by a study as follows [85]. The first issue is safety, since mRNA is a non-infectious, non-integrating platform, and shows no potential risk of infection or insertional mutagenesis. In addition, mRNA is degraded by regular cellular processes, and its half-life in vivo can be regulated through modifications and delivery methods. The second issue is its efficacy; modifications can make the mRNA more stable and highly translatable. Third, mRNA vaccines have the prospective for fast, inexpensive, and accessible manufacturing due to the high yields of in vitro transcription reactions. 

However, this platform is vulnerable in a high mutagenic potential virus [86]. Regarding the SARS-CoV-2 mutagenic potential, a study by Phan [87] revealed many mutations and deletions in coding and non-coding regions, mainly associated with the S protein. Another study [88] identified 11 variations in the SARS-CoV-2 genome which were observed in over 10% of patient isolates from all over the world. Therefore, this topic is still controversial. Nonetheless, this vaccine platform, despite needing adjuvants (other technologies), is more agile and scalable than others. In the selected CTPs, there are seven types of mRNA vaccines: four named BNT162 [14,15,18,19] and three with mRNA1273 [17,20,55], the latter being at a more advanced stage of development. Both types of vaccine consist of LNP encapsulated nucleoside modified mRNA targeting the S protein of SARS-CoV-2. The mRNA-1273 developed by the National Institute of Allergy and Infectious Disease (NIAID) in the USA in collaboration with ModernaTX Inc. [17,20,55] produces the mRNA of S protein in a stable form due to advances in carriers such as LNPs, providing high-efficiency delivery. 

Bacteria-derived plasmid-DNA vaccines, in addition to encoding the target antigen, can also encode co-stimulating molecules. These are directed to the nucleus, and need to cross the plasma and nuclear membranes to be activated. Furthermore, they have the possible disadvantage of persistent genomic integration in host cell DNA, leading to subsequent deleterious effects. However, the need for specific delivery systems to achieve good immunogenicity remains a concern. Different methods have been developed to enhance DNA uptake, expression, and immunogenicity (i.e., encapsulation in containing cationic lipids or cholesterol nanoparticles, and adsorption to polymeric or biodegradable nanoparticles), such as the gene gun, needle-free injection devices (jet injection) [89], and in vivo electroporation. This is the case for the INO-4800 vaccine against SARS-CoV-2, currently in clinical trial (phase I) [27], which was administered by intradermal (ID) injection followed by electroporation (EP) using a CELLECTRA^®^ 2000 device in healthy adult volunteers [27]. The synthesis process of this vaccine involved the alignment of four coding sequences of S protein and the addition of the N-terminal IgE leader sequence. This highly optimized DNA sequence was manufactured and cloned into pGX0001 vector. The resulting plasmids were named pGX9501 and pGX9503 by Inovio Pharmaceuticals [90]. The immunogenicity of INO-4800 was pre-clinically evaluated, and the serum reactivity revealed IgG binding against the S protein of SARS-CoV-2. The serum from INO-4800-immunized BALB/c mice neutralized two SARS-CoV-2 strains—WH-09 (Neutralizing Titer of Antibody (ND50: 97.5) and VIC01 (ND50: 128.1)). In addition, the ND50 of 573.5 was observed in immunized guinea pigs. This study also generated robust S-specific T cell responses in these models, and the detected antibodies were able to block S protein binding to the host ACE2 receptor [90].

Recently (17 June 2020), another DNA vaccine developed in South Korea by Genexine Consortium entered phase I-II clinical trials, in which a preventive GX-19 vaccine will be intramuscularly administrated in healthy volunteers from countries such as Indonesia and Thailand. The company managers expect preliminary data from the initial trial by September 2020 and hope to complete all phases of testing by the end of 2021 [25]. To date, no preclinical studies or technical specifications for the GX-19 vaccine have been released. Hence, DNA and mRNA vaccines are readily designed and can proceed into clinical trials very fast, in addition to outstanding targets for the development of vaccines against SARS-CoV-2 and other related epidemics in the future [91]. 

Attenuated or inactivated vaccines, known as conventional vaccines, require whole pathogen cultivation and propagation. Thus, it is necessary to obtain a high biosafety level with specialized laboratories and biotechnological tools. Moreover, there is the requirement for lineage cells accepted by regulatory authorities, such as Vero Cells, for the development of industrial-scale inactivated virus vaccines [92]. A total of 18% of the CTPs use inactivated virus vaccine platforms, including a phase III CTP recruiting in Abu Dhabi/the United Arab Emirates [53], and a phase III CTP in Brazil [54]. This approach has been used previously and led to smallpox eradication [93], and vaccines for other diseases such as polio, tetanus, diphtheria, and measles. The clinical trials developed by Sinovac Corporation [10] cultivated the CN2 strain of SARS-CoV-2 in Vero Cells and chemically inactivated them using β-propiolactone. Formaldehyde and UV light are other possible agents for virus inactivation [92]. The process of viral inactivation is delicate and cannot destroy the major virus antigenic structures, which would interfere with their immunogenicity. Depth filtration and optimized steps of chromatography allow vaccine purification [69]. Prior to CTPs, PiCoVacc initiated pre-clinical experiments and administered them with alum adjuvant in BALB/C mice and rhesus macaques, a non-human primate species. This study demonstrated rapid RBD-IgG development, accounting for half of the S protein-induced antibodies produced and possibly the dominant immunogenic part of this protein [69]. Furthermore, no antibody-dependent enhancement (ADE)-mediated vaccine-induced infection aggravation was observed for any vaccinated rhesus macaques [69]. Two other CTPs developed by Chinese Sinopharm Corporation at the Wuhan [8] and Beijing [9] Institutes of Biological Products also expanded SARS-CoV-2 in Vero Cells and are currently in trial. However, no pre-clinical studies or technical specifications of the last two vaccines have been disclosed to date. 

Since protein subunit vaccines are restricted to specific epitopes of the virus, most developers have not found proteins other than the S protein used by SARS-CoV-2 to invade cells [94]. Eight clinical trial protocols are analyzing protein subunit vaccines [37,38,39,40,41,42,43,44] combined with different types of adjuvants to help individuals produce an immune response strong enough to protect them from the disease [95]. However, the produced immune responses weaken over time, which means that an individual may require additional shots for booster immunizations throughout their life [94]. The adjuvants used in the selected clinical trial protocols are licensed and approved by the Food and Drug Administration (FDA). The adjuvant CpG1018 increases the body’s immune response; the AS03 enhances the vaccine antigen-specific adaptive immune response; and potassium aluminum sulfate (Alum), one of the agents most used as an adjuvant, acts by creating a deposit at the injection site, thus allowing the slow release of antigen and prolonging the interaction time with APCs, in addition to acting on soluble antigens by converting them into particulate forms that are readily phagocyted [95,96]. 

The global vaccine R&D efforts against SARS-CoV-2 are unprecedented in terms of scale and speed [73]. From examining all the CTPs operating in the United States (clinicaltrials.gov), China (chictr.org.cn), and Europe (clinicaltrialsregister.eu), it is evident that most COVID-19 vaccine CTP activity is taking place in China, with 19 CTPs (38%). However, the greatest advance has been made in the United Kingdom (Oxford University), in conjunction with several collaborating countries (South Africa and Brazil, among others) and other recruiting centers/institutions in the UK (hospitals, universities, labs, etc.). This CTP has reached phase III protocol [50], and will test 12,000 subjects. On 26 July 2020, its performance was 18.6%. Most CTPs (91.2%) have not collaborated with other external institutions due to the protocol phase; only 6 of 50 (12%) CTPs have reached phase III, for which it is necessary to expand the tests in territories with the outbreak, as seen with the Oxford University protocol, and more recently with Sinovac Life Sciences Co., a Chinese vaccine company. The distribution of collaboration networks between sponsors and the recruiting centers/institutions of the CTPs is strongly concentrated in Asia (23.2%), particularly in Wuhan city, China, where the pandemic started. In Europe and the Americas, the United Kingdom, Brazil, and the USA are the main centers, representing 82% of the global vaccine network or total subjects in vaccine tests. As we discussed earlier, these last countries are now the epicenter of the pandemic. Furthermore, the development of vaccine platforms is occurring more rapidly in wide global networks compared to scientific reports or clinical trial databases. Vaccine developers publish very few clinical trial articles, even regarding the safety or preclinical results.

The pandemic caused by COVID-19 has resulted in increased awareness of global threats to human health, particularly when caused by unknown and emerging pathogens. This pandemic has motivated science and health systems to be prepared for future outbreaks. In addition, global development has begun on vaccine platforms and improvements in public and private health systems to tackle the challenges of new outbreaks. New platforms or approaches, such as viral vectors, protein-like vaccines, and nucleic acids, meet the prerequisites for providing solutions to some of these challenges, representing highly versatile technologies that allow rapid vaccine manufacturing. Each vaccine technology has its own advantages and disadvantages related to its ability to induce certain immune responses, manufacturing capacity, and safety for human use.

## 5. Conclusions

The race of COVID vaccine remains uncertain, but to date global labs and industries have registered 50 vaccine CTPs in leading research databases, using eight platforms: inactivated virus; nucleic acid, such as mRNA, and DNA/plasmid; recombinant virus vectors; recombinant viral protein subunits; modified dendritic cells; artificial antigen-presenting cell; and VLP vaccine. New groups and laboratories have changed the global vaccine development landscape, especially in Asia. The most advanced Trial does not mean to be the most efficient or safe, the collaborations between countries are still of little significance. 

## Figures and Tables

**Figure 1 vaccines-08-00474-f001:**
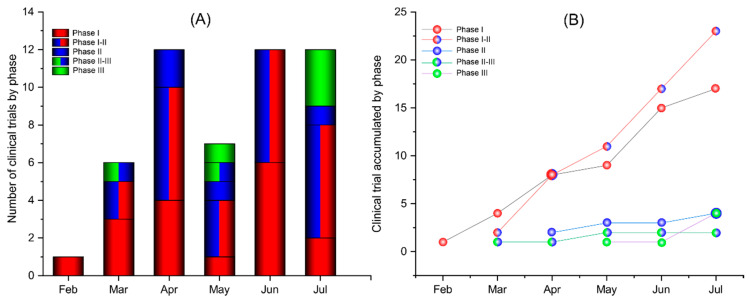
Over-time distribution of multiplatform vaccine clinical trials according to the study phase. (**A**) Clinical trial protocols beginning in each month classified by the phase of development. (**B**) Cumulative analysis of clinical trial phases over the months.

**Figure 2 vaccines-08-00474-f002:**
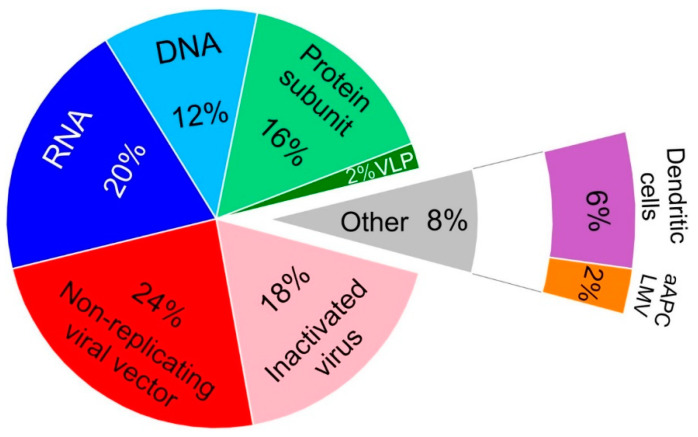
Clinical trial protocols multiplatform distribution for the development of vaccine candidates based on nucleic acid platforms (blue colors) comprising RNA and DNA vaccines, non-replicating viral vectors (**red**), inactivated SARS-CoV-2 (**pink**), protein subunits (**light green**), virus-like particles (**dark green**), dendritic cells (**purple**), and artificial antigen presentation cells modified by lentiviral vector (**orange**). Abbreviations: VLP: Virus-like particle; aAPC: artificial antigen presentation cells; LMV: lentiviral modified vector.

**Figure 3 vaccines-08-00474-f003:**
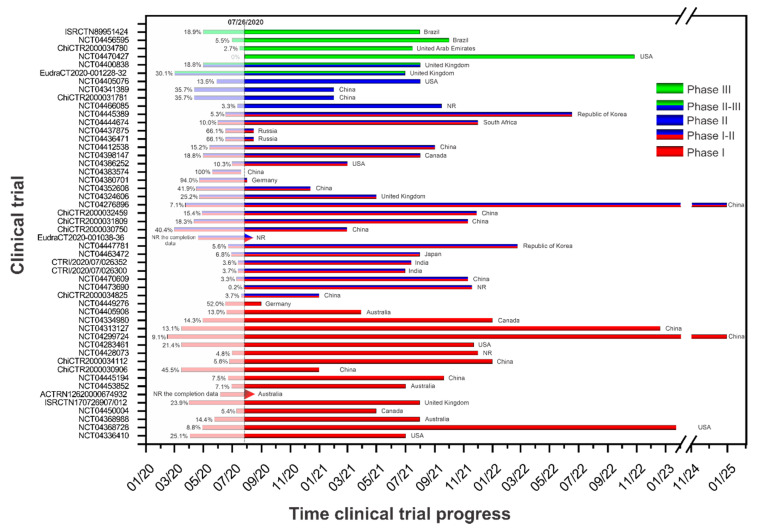
Timing clinical trial progress analysis by the phase of study of each clinical trial protocol for active immunization against COVID-19 (vaccine) and the timing progress rate (TPR) according to the initiation and completion date of each study. Note: * the NCT04324606 clinical trial protocol was also registered in the European database for clinical studies with the following identification: EudraCT 2020-001072-15.

**Figure 4 vaccines-08-00474-f004:**
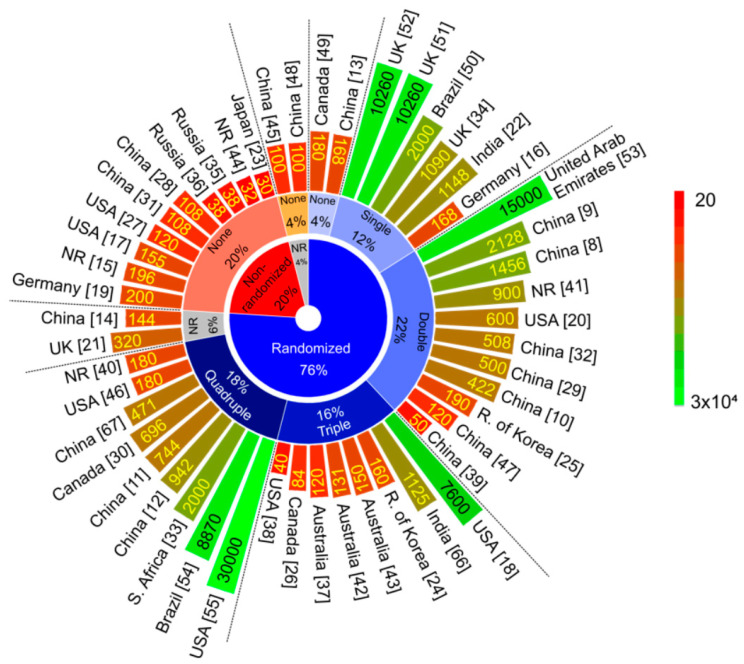
Study design of vaccine clinical trials against COVID-19 distributed inside out by the different types of allocation, masking, estimated enrollment, rehearsal center, and study countries. The color scale bar represents the number of volunteers estimated in each protocol. Abbreviations: NR—not reported; UK—the United Kingdom; USA—the United States of America.

**Figure 5 vaccines-08-00474-f005:**
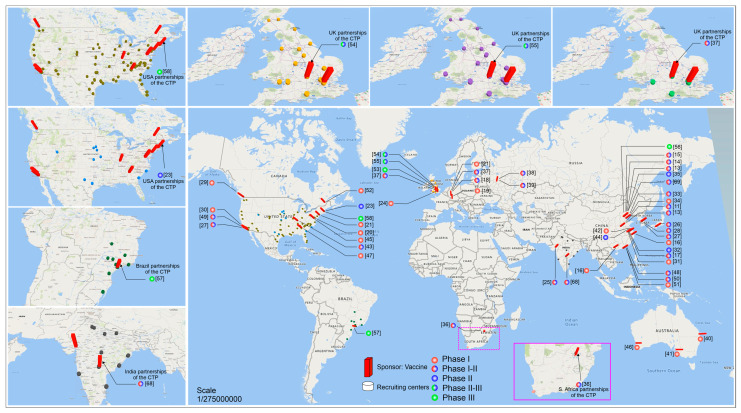
The global distribution of clinical trials by phase (**circles**) that are developing research on vaccines against COVID-19 (**red bar**) in the world and their recruitment centers (**cylinder**). This map displays only the centers that had more than 5 recruitment centers. In particular, the main centers of each continent are arranged around the central map. Phase I (**red circle**), phase I-II (**red-blue circle**), phase II (**blue circle**), phase II-III (**green-blue circle**), and phase III (**green circle**).

**Figure 6 vaccines-08-00474-f006:**
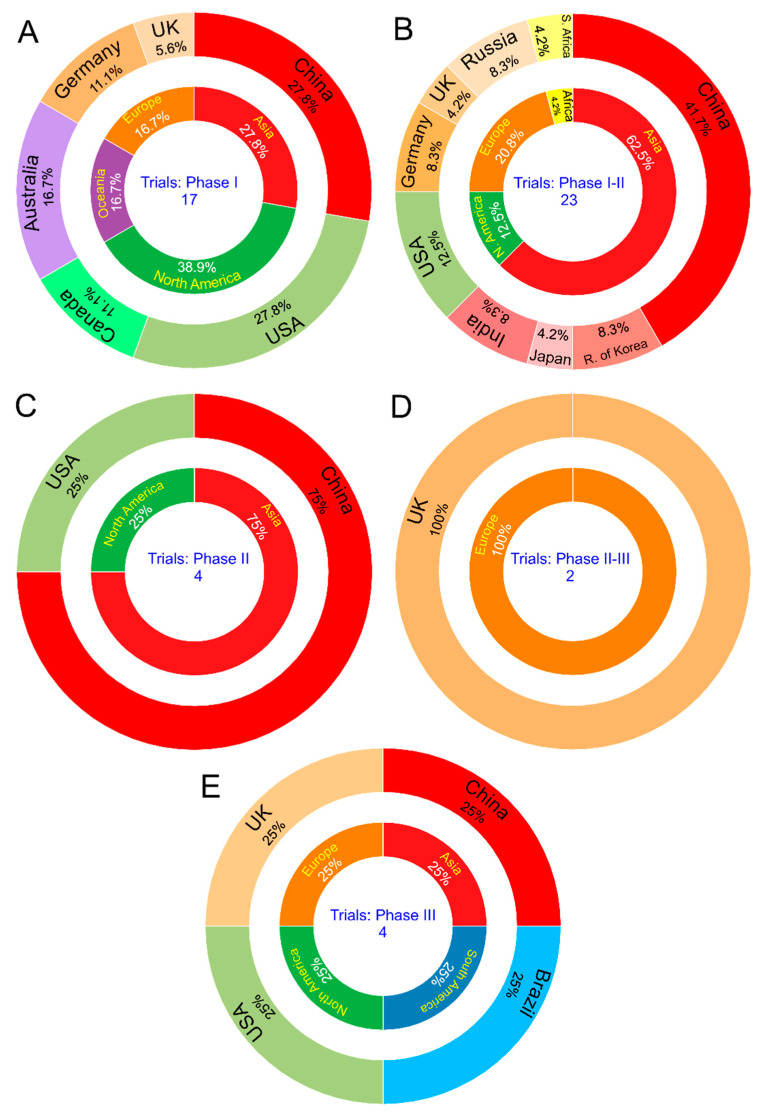
Clinical trial distribution by country/continent and phase. (**A**): CTPs at phase I; (**B**): CTPs at phase I-II; (**C**): CTPs at phase II; (**D**): CTPs at phase II-III; and (**E**): CTPs at phase III.

**Table 1 vaccines-08-00474-t001:** Description of the trial vaccine.

ID Number	Phase	Vaccine Name	Properties	Vaccine Features	Coronavirus Target	Start Date	Completion Date	Progress (%)	Recruitment Status	Recruitment Country
ISRCTN89951424# [50]	III	ChAdOx1 nCoV-19	Non replicating viral vector	Chimpanzee r-ADV vaccine encoding S protein	S protein of SARS-CoV-2	05/01/2020	07/31/2021	18.9	Recruiting	Brazil
NCT04456595 [54]	III	Adsorbed inactivated SARS-CoV-2	Inactivated virus	Adsorbed SARS-CoV-2 (CN2 strain) vaccine inactivated by BPL	Multiple proteins of SARS-CoV-2	07/01/2020	10/01/2021	5.5	Not yet recruiting	Brazil
ChiCTR2000034780 [53]	III	Purified inactivated SARS-CoV-2	Inactivated virus	SARS-CoV-2 strain inactivated inside Vero Cells	Multiple proteins of SARS-CoV-2	07/16/2020	07/16/2021	2.7	Recruiting	United Arab Emirates
NCT04470427 [55]	III	mRNA1273	mRNA	LNP-encapsulated mRNA-1273 encoding S protein	S protein of SARS-CoV-2	07/27/2020	10/27/2022	0.0	Not yet recruiting	USA
NCT04400838 [51]	II-III	ChAdOx1 nCoV-19	Non replicating viral vector	Chimpanzee r-ADV vaccine encoding S protein	S protein of SARS-CoV-2	05/01/2020	08/01/2021	18.8	Not yet recruiting	United Kingdom
EudraCT2020-001228-32 ISRCTN90906759 [52]	II-III	ChAdOx1 nCoV-19	Non replicating viral vector	Chimpanzee r-ADV vaccine encoding S protein	S protein of SARS-CoV-2	03/02/2020	06/30/2021	30.1	Ongoing	United Kingdom
NCT04405076 [20]	II	mRNA1273	mRNA	LNP-encapsulated mRNA-1273 encoding S protein	S protein of SARS-CoV-2	05/29/2020	08/01/2021	13.5	Recruiting	USA
NCT04341389 [32]	II	Adenovirus Type 5 (Ad5-nCoV)	Non replicating viral vector	Serotype 5 r-ADV vaccine encoding S protein	Full-length S protein of SARS-CoV-2	04/12/2020	01/31/2021	35.7	Active, not recruiting	China
ChiCTR2000031781 [29]	II	Adenovirus Type 5 (Ad5-nCoV)	Non replicating viral vector	Serotype 5 r-ADV vaccine encoding S protein	Full-length S protein of SARS-CoV-2	04/12/2020	01/31/2021	35.7	Not yet recruiting	China
NCT04466085 [41]	II	CHO cells vaccine	Protein Subunit	Recombinant protein produced with CHO cells + adjuvant (RBD-Dimer)	S protein of SARS-CoV-2	07/12/2020	09/15/2021	3.3	Not yet recruiting	NR
NCT04445389 [25]	I-II	GX-19	DNA	DNA vaccine	Unspecified protein of SARS-CoV-2	06/17/2020	06/17/2022	5.3	Recruiting	Republic of Korea
NCT04444674 [33]	I-II	ChAdOx1 nCoV-19	Non replicating viral vector	Chimpanzee r-ADV vaccine encoding S protein	S protein of SARS-CoV-2	06/01/2020	12/01/2021	10.0	Not yet recruiting	South Africa
NCT04437875 [36]	I-II	Gam-COVID-Vac Lyo	Non replicating viral vector	Combined serotypes 5 and 26 r-ADV vectored vaccine encoding S protein	S protein of SARS-CoV-2	06/17/2020	08/15/2020	66.1	Recruiting	Russia
NCT04436471 [35]	I-II	Gam-COVID-Vac	Non replicating viral vector	Combined serotypes 5 and 26 r-ADV vectored vaccine encoding S protein	S protein of SARS-CoV-2	06/17/2020	08/15/2020	66.1	Recruiting	Russia
NCT04412538 [12]	I-II	Purified inactivated SARS-CoV-2	Inactivated virus	Purified inactivated SARS-CoV-2	Multiple proteins of SARS-CoV-2	05/15/2020	09/01/2021	15.2	Recruiting	China
NCT04398147 [30]	I-II	Adenovirus Type 5 (Ad5-nCoV)	Non replicating viral vector	Serotype 5 r-ADV vaccine encoding S protein intramuscularly	Full-length S protein of SARS-CoV-2	05/01/2020	08/01/2021	18.8	Not yet recruiting	Canada
NCT04386252 [46]	I-II	AV-COVID-19	Dendritic cells	Autologous DCs differentiated in vitro from monocytes incubated with IL-4 and GM-CSF loaded with antigens from SARS-CoV-2	Unspecified proteins of SARS-CoV-2	07/01/2020	03/01/2021	10.3	Not yet recruiting	USA
NCT04383574 [10]	I-II	Adsorbed inactivated SARS-CoV-2	Inactivated virus	Adsorbed SARS-CoV-2 (CN2 strain) vaccine inactivated by BPL	Multiple proteins of SARS-CoV-2	05/20/2020	07/20/2020	100	Completed	China
NCT04380701 [19]	I-II	BNT162	mRNA	LNP-encapsulated nucleoside modified mRNA (BNT162)	S protein of SARS-CoV-2	04/23/2020	08/01/2020	94.0	Recruiting	Germany
NCT04352608 [11]	I-II	Adsorbed inactivated SARS-CoV-2	Inactivated virus	Adsorbed SARS-CoV-2 (CN2 strain) vaccine inactivated by BPL	Multiple proteins of SARS-CoV-2	04/16/2020	12/13/2020	41.9	Recruiting	China
NCT04324606 EudraCT2020-001072-15 [34]	I-II	ChAdOx1 nCoV-19	Non replicating viral vector	Chimpanzee r-ADV vaccine encoding S protein	S protein of SARS-CoV-2	04/23/2020	05/01/2021	25.2	Active, not recruiting	United Kingdom
NCT04276896 [45]	I-II	Lentiviral Minigene vaccine (LV-SMENP)	Dendritic cells	DCs modified by lentiviral vector system (NHP/TYF) + CTLs	Multiple proteins of SARS-CoV-2	03/24/2020	12/31/2024	7.1	Recruiting	China
ChiCTR2000032459 [9]	I-II	Purified inactivated SARS-CoV-2	Inactivated virus	SARS-CoV-2 strain inactivated inside Vero Cells	Multiple proteins of SARS-CoV-2	04/28/2020	11/28/2021	15.4	Recruiting	China
ChiCTR2000031809 [8]	I-II	Purified inactivated SARS-CoV-2	Inactivated virus	SARS-CoV-2 strain inactivated inside Vero Cells	Multiple proteins of SARS-CoV-2	04/11/2020	11/10/2021	18.3	Not yet recruiting	China
ChiCTR2000030750 [47]	I-II	Dendritic cells vaccine	Dendritic cells	COVID-19 epitope gene recombinant chimeric DC vaccine	SARS-CoV-2 epitope	03/01/2020	02/28/2021	40.4	Not yet recruiting	China
EudraCT 2020-001038-36 [15]	I-II	BNT162	mRNA	LNP-encapsulated nucleoside modified mRNA (BNT162)	S protein epitope of SARS-CoV-2	04/20/2020	07/26/2020 *	NR	Ongoing	NR
NCT04447781 [24]	I-II	INO-4800	DNA	DNA plasmid (pGX9501) vaccine with electroporation (INO-4800)	Full-length S protein of SARS-CoV-2	06/22/2020	02/22/2022	5.6	Not yet recruiting	Republic of Korea
NCT04463472 [23]	I-II	AG0301-COVID19	DNA	DNA plasmid vaccine	Multiples antigens from SARS-CoV-2	06/29/2020	07/31/2021	6.8	Recruiting	Japan
CTRI/2020/07/026352 [22]	I-II	ZYCOV-D	DNA	DNA plasmid vaccine	S protein of SARS-CoV-2	07/13/2020	07/13/2021	3.6	Recruiting	India
CTRI/2020/07/026300/NCT04471519 [66]	I-II	Covaxin (BBV152)	Inactivated virus	Whole-Virion Inactivated SARS-CoV-2 Vaccine (BBV152A, BBV152B and BBV152C)	Multiples antigens from SARS-CoV-2	07/13/2020	06/30/2021	3.7	Recruiting	India
NCT04470609 [67]	I-II	SARS-CoV-2 Vaccine	Inactivated virus	Purified inactivated SARS-CoV-2 vaccine	Multiples antigens from SARS-CoV-2	07/10/2020	11/10/2021	3.3	Enrolling by invitation	China
NCT04473690 [40]	I-II	KBP-COVID-19	Protein Subunit	RBD-based vaccine developed with fast-growing tobacco plant technology	RBD S protein of SARS-CoV2	07/25/2020	11/18/2021	0.2	Not yet recruiting	NR
ChiCTR2000034825 [14]	I-II	BNT162b1	mRNA	3 LNP-mRNAs	RBD S protein of SARS-CoV2	07/20/2020	12/31/2020	3.7	Recruiting	China
NCT04449276 EudraCT 2020-001286-36 [16]	I	CVnCoV	mRNA	mRNA vaccine	Unspecified protein of SARS-CoV-2	06/17/2020	08/31/2020	52.0	Recruiting	Germany
NCT04405908 [43]	I	SCB-2019	Protein subunit	Recombinant 2019-nCoV S protein subunit-trimer vaccine + AS03 or CpG 1018 + Alum adjuvants	S protein of SARS-CoV-2	06/19/2020	03/30/2021	13.0	Recruiting	Australia
NCT04334980 [26]	I	bacTRL-Spike	DNA	Genetically modified probiotic bacteria containing plasmid encoding S protein	S protein of SARS-CoV-2	04/30/2020	12/31/2021	14.3	Not yet recruiting	Canada
NCT04313127 [31]	I	Adenovirus Type 5 (Ad5-nCoV)	Non replicating viral vector	r-ADV vaccine encoding S protein	Full-length S protein of SARS-CoV-2	03/16/2020	12/20/2022	13.1	Active, not recruiting	China
NCT04299724 [48]	I	aAPC	aAPC lentiviral modified vector	aAPC modified by lentiviral vector system NHP/TYF	Multiple proteins of SARS-CoV-2	02/15/2020	12/31/2024	9.1	Recruiting	China
NCT04283461 [17]	I	mRNA-1273	mRNA	LNP-encapsulated mRNA-1273	S protein of SARS-CoV-2	03/16/2020	11/22/2021	21.4	Recruiting	USA
NCT04428073 [44]	I	Covax-19™	Protein subunit	Advax™ adjuvant with a recombinant SARS-CoV-2 S protein	S protein of SARS-CoV-2	07/01/2020	12/01/2021	4.8	Not yet recruiting	NR
ChiCTR2000034112 [13]	I	NR	mRNA	mRNA vaccine	RBD S protein of SARS-CoV2	06/25/2020	12/31/2021	5.6	Not yet recruiting	China
ChiCTR2000030906 [28]	I	Adenovirus Type 5 (Ad5-nCoV)	Non replicating viral vector	Serotype 5 r-ADV vaccine encoding S protein intramuscularly	Full-length S protein of SARS-CoV-2	03/16/2020	12/31/2020	45.5	Recruiting	China
NCT04445194 [39]	I	CHO cells vaccine	Protein Subunit	Adjuvanted recombinant protein (RBD-Dimer)	S protein of SARS-CoV-2	06/22/2020	09/20/2021	7.5	Recruiting	China
NCT04453852 [38]	I	Covax-19™	Protein Subunit	Advax™ adjuvant combined with a SARS-CoV-2 recombinantS protein	S protein of SARS-CoV-2	06/30/2020	07/01/2021	7.1	Recruiting	Australia
ACTRN12620000674932 [37]	I	SARS-CoV-2 Sclamp	Protein Subunit	Molecular clamp stabilized S protein with MF59 adjuvant	S protein of SARS-CoV-2	06/06/2020	07/26/2020 *	NR	Recruiting	Australia
ISRCTN17072692 [21]	I	LNP-nCoVsaRNA	saRNA	saRNA vaccine encoding S protein	S protein of SARS-CoV-2	04/01/2020	07/31/2021	23.9	Recruiting	United Kingdom
NCT04450004 [49]	I	Plant-derived VLP	VLP	Plant-derived VLP + CpG 1018 or AS03 adjuvants	S protein of SARS-CoV-2	07/10/2020	04/30/2021	5.4	Recruiting	Canada
NCT04368988 [42]	I	NVX-CoV2373	Protein subunit	Recombinant SARS CoV-2 S protein NP vaccine + Matrix M adjuvant	S protein of SARS-CoV-2	05/25/2020	07/31/2021	14.4	Recruiting	Australia
NCT04368728 [18]	I	BNT162	mRNA	LNP-encapsulated nucleoside modified mRNA (BNT162)	S protein epitope of SARS-CoV-2	04/29/2020	01/23/2023	8.8	Recruiting	USA
NCT04336410 [27]	I	INO-4800	DNA	DNA plasmid (pGX9501) vaccine with electroporation (INO-4800)	Full-length S protein of SARS-CoV-2	04/03/2020	07/01/2021	25.1	Recruiting	USA

**Abbreviations**: mRNA: messenger ribonucleic acid; Ad5-nCoV: recombinant adenovirus type 5; r-ADV: recombinant adenovirus-vectored; CHO: Hamster Ovary Cell; CTL: antigen-specific cytotoxic T cell; LV-SMENP: modifying Dendritic Cell with lentivirus vectors expressing COVID-19 minigene SMENP; DC: Dendritic Cell; KBP: Kentucky BioProcessing; SCB-2019: S-Trimer COVID-19 Vaccine; aAPC: artificial antigen presenting cells; NR: Not Reported; LNP: Lipid Nanoparticle; NP: nanoparticle; VLP: Virus Like Particle; INO-4800: INOVIO’s DNA vaccine candidate; DNA: deoxyribonucleic acid; S protein: Spike protein; CN2: strain CN2 for purified inactivated SARS-CoV-2 virus vaccine; RBD: receptor-binding domain; Ad26: recombinant adenovirus type 26; BPL: β-propiolactone GM-CSF: granulocyte-macrophage colony-stimulating factor; NHP/TYF: self-inactivating lentiviral vector system; AS03: squalene-based immunologic adjuvant; CpG 1018: cytosine phosphoguanine 1018—the adjuvant contained in HEPLISAV-B^®^; Alum: potassium aluminum sulfate; MF59: immunologic adjuvant that uses squalene; saRNA: Self-amplifying ribonucleic acid; USA: United States of America. Note: # This clinical trial has a partnership with the study NCT04400838. * These clinical trials did not mention the study completion date, so we have not calculated the progression rate of these studies.

**Table 2 vaccines-08-00474-t002:** Study design, arms, and interventions.

ID Number	Rehearsal Center	Estimated Enrollment	Allocation	Intervention Model	Masking	Intervention (Route)	Arm	Dose (Day)	Age Range (Years)
ISRCTN89951424 [50]	Single center	2000	Randomized	Sequential Assignment	Single	ChAdOx1 nCoV-19 (i.m) or MenACWY (i.m)	2	ChAdOx1 nCoV-19: 5 × 10^10^ vp Men ACWY: 0.5 ml	18–55
NCT04456595 [54]	Multicenter	8870	Randomized	Parallel Assignment	Quadruple	Inactivated SARS-CoV-2 Vaccine (i.m. − deltoid muscle) × Placebo (i.m)	4	NR (2 times: 0,14)	>18
ChiCTR2000034780 [53]	Single center	15,000	Randomized	Parallel Assignment	Double	Inactivated SARS-CoV-2 Vaccine × Placebo	3	NR (2 times)	>18
NCT04470427 [55]	Multicenter	30,000	Randomized	Parallel Assignment	Quadruple	mRNA-1273 Vaccine (.i.m) × Placebo (i.m)	2	100 μg (2 times: 0,28)	>18
NCT04400838 [51]	Multicenter	10,260	Randomized	Sequential Assignment	Single	ChAdOx1 nCoV-19 (i.m) × MenACWY vaccine (licensed control vaccine (i.m)	14	2.5 × 10^10^ vp; 5 × 10^10^ vp (single or 2 times: 1,28)	>5
EudraCT2020-001228-32 ISRCTN90906759 [52]	Multicenter	10,260	Randomized	NR	Single	ChAdOx1 nCoV-19 (i.m) × MenACWY vaccine (licensed control vaccine (i.m)	5	NR	5–12; 18–55; >56
NCT04405076 [20]	Multicenter	600	Randomized	Sequential Assignment	Double	Crossover: mRNA-1273 SARS-COV-2 Vaccine <---> Placebo	4	50 mcg; 100 mcg (0)	18–54; >55
NCT04341389 [32]	Single center	508	Randomized	Crossover Assignment	Double	rAd5-nCoV (i.m) × Placebo (i.m)	3	1 × 10^11^vp; 5 × 10^10^ vp (0)	>18
ChiCTR2000031781 [29]	Multicenter	500	Randomized	Parallel Assignment	Double	rAd5-nCoV (i.m) × Placebo (i.m)	3	5 × 10^10^vp; 1 × 10^11^ vp	>18
NCT04466085 [41]	NR	900	Randomized	Parallel Assignment	Double	Recombinant new coronavirus vaccine (CHO cells) (i.m.− deltoid muscle) × Placebo (i.m − deltoid muscle)	6	25 μg/0.5 mL; 50 μg/0.5 mL (2 and 3 times: 0,1 month)	18–59
NCT04445389 [25]	Single center	190	Randomized	Parallel Assignment	Double	GX-19 (i.m) × Placebo (i.m)	3	GX-19 dose A; B (1, 29)	18–50
NCT04444674 [33]	Multicenter	2000	Randomized	Parallel assignment	Quadruple	ChAdOx1 nCoV-19 (i.m. − deltoid muscle x Placebo (i.m. − deltoid muscle)	8	5 × 10^10^ vp (single or 2 times; 0,28)	18–65
NCT04437875 [36]	Single center	38	Non-randomized	Parallel Assignment	None	rAd26 (i.m) × rAd5 (i.m.) × rAd26 + rAd5 (i.m.)	3	rAd26 (1), rAd5 (1), rAd26 (1) + rAd5 (21)	18–60
NCT04436471 [35]	Single center	38	Non-randomized	Parallel Assignment	None	rAd26 (i.m) × rAd5 (i.m.) × rAd26 + rAd5 (i.m.)	3	rAd26 (1), rAd5 (1), rAd26 (1) + rAd5 (21)	18–60
NCT04412538 [12]	Single center	942	Randomized	Parallel Assignment	Quadruple	Inactivated SARS-CoV-2 Vaccine × Placebo	8	50 U/0.5 mL; 100 U/0.5 mL; 150 U/0.5 mL (2 times: 0,14 or 0,28)	18–59
NCT04398147 [30]	Single center	696	Randomized	Parallel Assignment	Quadruple	rAd5-nCoV (i.m) × Placebo (i.m)	28	5 × 10^10^ vp; 10 × 10^10^ vp (single or 2 times: 0,56)	18–55;65–85;
NCT04386252 [46]	Single center	180	Randomized	Parallel Assignment	Quadruple	Dendritic cells vaccine × GM-CSF (s.c) × Placebo (s.c)	9	DC loaded with 1×, 10× and 30× antigen + 250 and 500 mcg GM-CSF (0)	>18
NCT04383574 [10]	Single center	422	Randomized	Parallel Assignment	Double	Inactivated SARS-CoV-2 vaccine × Placebo	4	300 SU/mL; 600 SU/mL; 1200 SU/mL (2 times: 0,28)	>60
NCT04380701 [19]	Single center	200	Non-randomized	Sequential Assignment	None	BNT162a1; BNT162b1; BNT162b2; BNT162c2	4	Escalating dose levels (BNT162a1, BNT162b1, BNT162b2); Single dose (BNT162c2)	18–55
NCT04352608 [11]	Single center	744	Randomized	Parallel Assignment	Quadruple	Inactivated SARS-CoV-2 vaccine (i.m) × Placebo (i.m.)	6	600 SU/0.5 mL; 1200 SU/0.5 mL (2 times: 0,14 or 0,28)	18–59
NCT04324606 EudraCT2020-001072-15 [34]	Multicenter	1090	Randomized	Sequential Assignment	Single	ChAdOx1 nCoV-19 (i.m) × MenACWY vaccine (i.m) ± Paracetamol (oral)	9	5 × 10^10^ vp (Single: 0)	18–55
NCT04276896 [45]	Multicenter	100	NR	Single Group Assignment	None	LV-SMENP-DC vaccine (s.c) and antigen-specific CTLs (i.v)	1	5 × 10^6^ DC + 1 × 10^8^ CTLs	0.5–80
ChiCTR2000032459 [9]	Single center	2128 (I: 480; II: 1648)	Randomized	Parallel Assignment	Double	Inactivated SARS-CoV-2 vaccine × Placebo	30 (Phase I); 38 (Phase II)	Low; Medium; High	>3
ChiCTR2000031809 [8]	Multicenter	1456 (I: 288; II: 1168)	Randomized	Parallel Assignment	Double	Inactivated SARS-CoV-2 vaccine × Placebo	18 (Phase I); 26 (Phase II)	Low; Medium; High	>6
ChiCTR2000030750 [47]	Multicenter	120	Randomized	Parallel Assignment	Double	Recombinant chimeric DC vaccine × Blank vaccine	4	NR	25–65
EudraCT 2020-001038-36 [15]	Multicenter	196	Non-randomized	Parallel Assignment	None	BNT162a1 × BNT162b1 × BNT162b2 × BNT162c2	4	Prime/Boost Regimen	18–64
NCT04447781 [24]	Single center	160	Randomized	Sequential Assignment	Triple	INO-4800 (i.d. + EP) × Placebo (i.d. + EP)	4	1 mg/dose + EP; 2 mg/dose + EP (2 times: 0,28)	19–64
NCT04463472 [23]	Single center	30	Non-Randomized	Sequential Assignment	None	AG0301 DNA Vaccine (i.m.)	2	1.0 mg; 2.0 mg (2 times: 0,14)	20–65
CTRI/2020/07/026352 [22]	Single center	1148	Randomized	Sequential Assignment	Single	Novel Corona Virus-2019-nCov vaccine (i.d.) × Placebo (i.d)	NR	0.1 mL (Three times: 0,28,56)	18–55
CTRI/2020/07/026300 NCT04471519 [66]	Multicenter	1125	Randomized	Parallel Assignment	Triple	BBV152A (i.m) × BBV152B (i.m) × BBV152C (i.m.) × Placebo (i.m)	3	0.5 mL (Two times: 0;14)	12–55; 12–65
NCT04470609 [67]	Multicenter	471	Randomized	Parallel Assignment	Quadruple	Inactivated SARS-CoV-2 vaccine × Placebo	4	50 U/0.5 mL; 100 U/0.5 mL (2 times: 0,28)	>60
NCT04473690 [40]	NR	180	Randomized	Parallel Assignment	Quadruple	KBP × Placebo	3	Low and high doses	18–49; 50–70
ChiCTR2000034825 [14]	Multicenter	144	Randomized	Parallel Assignment	NR	BNT162b1 mRNA vaccine × Placebo	6	Low and high doses (2 times: 0,21)	18–55; >55
NCT04449276 EudraCT 2020-001286-36 [16]	Single center	168	Randomized	Sequential Assignment	Single	CVnCoV (i.m. − deltoid muscle) × Placebo (i.m − deltoid muscle)	2	2, 4 and 8 μg (1 and 29)	18–60
NCT04405908 [43]	Single center	150	Randomized	Sequential Assignment	Triple	SCB-2019 vaccine (i.m); SCB-2019 + AS03 (i.m) and SCB-2019 + CpG 1018 + Alum (i.m)	15	3 µg; 9 µg; 30 µg (1,22)	18–54; 55–75
NCT04334980 [26]	Multicenter	84	Randomized	Parallel Assignment	Triple	bacTRL-Spike (oral) × Placebo (oral)	6	1 cfu; 3 cfu; 10 cfu (0)	19–45
NCT04313127 [31]	Single center	108	Non-randomized	Sequential Assignment	None	Ad5-nCoV (i.m)	3	5 × 10^10^ vp; 1 × 10^11^ vp; 1.5 × 10^11^ vp (Single: 0)	18–60
NCT04299724 [48]	Single center	100	N/A	Single Group Assignment	None	Pathogen-specific aAPC vaccine (s.c)	1	5 × 10^6^ cells (Three times: 0,14,28)	0.5–80
NCT04283461 [17]	Multicenter	155	Non-randomized	Sequential Assignment	None	LNP- encapsulated mRNA-1273 (i.m.)	13	10 mcg; 25 mcg; 50 mcg; 100 mcg; 250 mcg (2 times; 1,29)	18–55; 56–70; >70
NCT04428073 [44]	Single center	32	Non-randomized	Sequential Assignment	None	Covax-19 vaccine	2	1.0 mL of low dose; 1.0 mL of high dose	18–60
ChiCTR2000034112 [13]	Multicenter	168	Randomized	Parallel Assignment	None	mRNA vaccine	3	Low; Medium; High	18–59; 60–80
ChiCTR2000030906 [28]	Multicenter	108	Non-randomized	Parallel Assignment	None	rAd5-nCoV (i.m.)	3	5 × 10^10^ vp; 1 × 10^11^ vp; 1.5 × 10^11^ vp	18–60
NCT04445194 [39]	Single center	50	Randomized	Parallel Assignment	Double	Recombinant new coronavirus vaccine (CHO cell) (i.m) × Placebo (i.m)	3	25 μg/0.5 mL; 50 μg/0.5 mL	18–59
NCT04453852 [38]	Single center	40	Randomized	Parallel Assignment	Triple	Covax-19 vaccine × Placebo	2	25 ug Spike antigen + 15 mg Advax-2 adjuvant	18–68
ACTRN12620000674932 [37]	Single center	120	Randomized	Parallel Assignment	Triple	SARS-CoV-2 Sclamp vaccine (i.m) × Placebo (i.m)	4	1 × 5 mcg/0.5 mL; 1 × 15 mcg/0.5 mL; 1 × 45 mcg/0.5 mL (2 times: 0,28)	18–55
ISRCTN17072692 [21]	Multicenter	320	Randomized	Sequential Assignment	NR	COVAC mRNA vaccine (i.m. − deltoid muscle)	3	0.1 µg; 0.3 µg; 1.0 µg	18–45; 18–75
NCT04450004 [49]	Multicenter	180	Randomized	Sequential Assignment	None	Coronavirus-like particle COVID-19 vaccine ± CpG 1018 or AS03 (i.m)	9	3.75 µg; 7.5 µg; 15 µg + CpG 1018 or AS03	18–55
NCT04368988 [42]	Multicenter	131	Randomized	Parallel Assignment	Triple	NVX-CoV2373 (Matrix-M) (i.m) × Placebo (i.m)	5	5 μg or 25 μg with/without 50 μg Matrix-M (2 times: 0,21)	18–59
NCT04368728 [18]	Multicenter	7600	Randomized	Parallel Assignment	Triple	BNT162a1 (i.m); BNT162b1 (i.m); BNT162b2 (i.m); BNT162c2 (i.m); Placebo (i.m)	21	0.5 mL (Single or 2 times: 0,21). Prime/boost regimen	18–55; 65–85; 18–85
NCT04336410 [27]	Multicenter	120	Non-randomized	Sequential Assignment	None	INO-4800 (i.d + EP)	3	0.5 mg + EP; 1.0 mg + EP; 2.0 mg + EP (2 times: 0,28)	>18

**Abbreviations:** NR: not reported; MenACWY: meningococcal ACWY licensed control vaccine; mRNA: messenger ribonucleic acid; CHO: Hamster Ovary Cell; rAd26: recombinant adenovirus serotype 26; rAd5-nCoV: recombinant adenovirus serotype 5; GM-CSF: Granulocyte Macrophage-colony Stimulating actor; LV-SMENP-DC: lentiviral minigene vaccine modified with DCs to activate T cells; DCs: dendritic cells; CTLs: cytotoxic T lymphocytes; INO-4800: INOVIO’s DNA vaccine candidate; KBP: Kentucky BioProcessing; AS03: squalene-based immunologic adjuvant made by GlaxoSmithKline (GSK); CpG 1018: cytosine phosphoguanine 1018; Alum: potassium aluminum sulfate; aAPC: artificial antigen presenting cells; LNP: lipid nanoparticle; Matrix-M: matrix-M ^TM^ adjuvant; i.m.: intramuscular; s.c.: subcutaneous; i.v.: intravenous; i.d.: intradermal; EP: electroporation.

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
