# Peer review of "Current Clinical Trials Protocols and the Global Effort for Immunization against SARS-CoV-2"

_vaccines, 2020, doi:10.3390/vaccines8030474_

Round 1
Reviewer 1 Report
Review article by Rego et al. (vaccines-896969) entitled “Current clinical trials protocols and the global effort for immunization against SARS-CoV-2” summarized recent trend on SARS-CoV-2 vaccine development. Their manuscript tried to comprehensively cover all vaccine trials and intended to provide perspectives on vaccine development. Such review article is currently very much in demand and it would fit the aim of the journal ”Vaccine”. Unfortunately, the text suffers many awkward and sometimes misleading wordings that requires major polishing. This reviewer feels that the manuscript should be significantly modified to mend sentences for publication in “Vaccine”.
Followings are the minor points need authors’ attention.
- Tables should be provided as supplemental information or reorganize to make them more presentable. Simple listing of such vast information is not appealing to readers. Summarized figure would be helpful.
- Many misplaced abbreviations; CTs, EUA(USA?).
Author Response
Reviewer #1
Comments and Suggestions for Authors
Review article by Rego et al. (vaccines-896969) entitled “Current clinical trials protocols and the global effort for immunization against SARS-CoV-2” summarized recent trend on SARS-CoV-2 vaccine development. Their manuscript tried to comprehensively cover all vaccine trials and intended to provide perspectives on vaccine development. Such review article is currently very much in demand and it would fit the aim of the journal ”Vaccine”. Unfortunately, the text suffers many awkward and sometimes misleading wordings that requires major polishing. This reviewer feels that the manuscript should be significantly modified to mend sentences for publication in “Vaccine”.
Answer: We have reviewed this manuscript in order to correct errors, misleading words and to improve the text general comprehension
Followings are the minor points need authors’ attention.
- Tables should be provided as supplemental information or reorganize to make them more presentable. Simple listing of such vast information is not appealing to readers. Summarized figure would be helpful.
Answer: Thank you for your suggestion. We improved the information in Tables 1 and 2 which contain the clinical trials core information for vaccine properties and study design Table 3 was transferred to supplementary information (Appendix A) according to your suggestion.
- Many misplaced abbreviations; CTs, EUA(USA?).
Answer: We have corrected the abbreviations in the manuscript
Reviewer 2 Report
This is a very detailed review of current vaccine development against SARS-CoV-2. It will be of tremendous interest to the field, given that there are so many different vaccines under development or in clinical trials.
Minor comments:
1) Line 63: what is CTPs? It is very confusing with line 81 where clinical trial protocols are labeled as “CPTs”.
2) The current table 1 does not follow any logical arrangement. I will suggest to group table 1 based on the vaccine properties. Additionally, several clinical trials listed in table 1 seems to use the same vaccine (e.g. ChAdOx1 nCoV-19). Perhaps it will be helpful to consider indicating/grouping which trials are using the same vaccine or carried out by the same university or company.
3) Line 198: what is BNT?
4) Section 3.3 is very fragmented with poor linkage between paragraphs. Therefore, I will suggest to add subtitles in section 3.3. to clearly separate each paragraph according to their topic. For example, RNA and DNA vaccines, inactivated vaccines etc. Additionally, the paragraph on virus like particles (VLP) vaccines is very lacking, with only 1 sentence. It either needs to be expanded or incorporated into other paragraphs.
5) Line 264 to 284 does not fit well with the rest of section 3.3 which was describing different types of vaccines. Perhaps line 264-284 should be in their own section or merge with the beginning of section 3.4.
6) Given that so many abbreviations (e.g. BNT) were used in this manuscript, please carefully check to ensure that their full words are clearly stated.
Author Response
Reviewer #2
Comments and Suggestions for Authors
This is a very detailed review of current vaccine development against SARS-CoV-2. It will be of tremendous interest to the field, given that there are so many different vaccines under development or in clinical trials.
Minor comments:
1) Line 63: what is CTPs? It is very confusing with line 81 where clinical trial protocols are labeled as “CPTs”.
Answer: We corrected the abbreviation at line 63 and added the meaning of the abbreviation. As also, we checked and corrected this abbreviation in all manuscript.
2) The current table 1 does not follow any logical arrangement. I will suggest to group table 1 based on the vaccine properties. Additionally, several clinical trials listed in table 1 seems to use the same vaccine (e.g. ChAdOx1 nCoV-19). Perhaps it will be helpful to consider indicating/grouping which trials are using the same vaccine or carried out by the same university or company.
Answer: Thank you for your suggestion. All tables are arranged by the clinical trial phases, from the most advanced studies (phase III) to the initial studies (phase I). The information about the group, university or company of vaccine development is highlighted in table A1 (Appendix A).
3) Line 198: what is BNT?
Answer: We added the meaning of BNT abbreviation to the manuscript
4) Section 3.3 is very fragmented with poor linkage between paragraphs. Therefore, I will suggest to add subtitles in section 3.3. to clearly separate each paragraph according to their topic. For example, RNA and DNA vaccines, inactivated vaccines etc. Additionally, the paragraph on virus like particles (VLP) vaccines is very lacking, with only 1 sentence. It either needs to be expanded or incorporated into other paragraphs.
Answer: We separated the types of vaccine by topic as requested by the reviewer. In addition, we added more information about the vaccine like particle virus (VLP), improving the comprehension for the vaccine type section.
5) Line 264 to 284 does not fit well with the rest of section 3.3 which was describing different types of vaccines. Perhaps line 264-284 should be in their own section or merge with the beginning of section 3.4.
Answer: Thank you for your suggestion. We agreed with the reviewer and created a separate section 3.4 for these two paragraphs.
6) Given that so many abbreviations (e.g. BNT) were used in this manuscript, please carefully check to ensure that their full words are clearly stated.
Answer: The abbreviations were corrected and standardized as requested.
Reviewer 3 Report
The manuscript number 896969, entitled “Current clinical trials protocols and the global effort for immunization against SARS-CoV-2” presents a systematic review summarizing the global race for vaccine development against COVID-19 based on 50 current studies selected from the main clinical trial databases. The authors also give a global perspective of the evolution, advantages and limitations of the eight vaccine platforms against COVID-19 under clinical trial protocols based on inactivated virus; on nucleic acid as mRNA, self-amplifying RNA (saRNA) and DNA/plasmid; recombinant adenovirus serotypes platforms as adenoviral vector 5, chimpanzee adenoviral vector ChAdOx1 and combined serotypes vectors Adenovirus 5 and 26; recombinant viral protein subunits modified dendritic cells; artificial antigen-presenting cells and virus like particles. In my opinion, this manuscript is adequate for Vaccine journal since it summarizes relevant information and gives a perspective of the global efforts under development to reach as soon as possible a vaccine candidate against COVID-19, the biggest health challenge of the 21st century that affects millions of people globally. However, some points should be addressed and clarified before its publication, especially in the introduction section.
- In line 40, the authors claim there is not consolidated therapeutic drug strategy approved for COVID-19. Then, it is briefly described the use of convalescent plasma antibodies. It can also be interesting to have a critical view about some approaches suggested as potential to treat or prevent COVID-19, such as the use of chloroquine or BCG vaccine. What happens with these approaches? Could they be efficient or not?
- Some problems of format should be improved, such as the lack of space before some references, the designation of the abbreviation CTPs should appear in line 63…
- I feel the lack of some information to compare all the clinical trials under development. Some of this information is appearing throughout the manuscript, mainly in the discussion section, but in my point of view, the best way to summarize and compare all the approaches is to include also in Table 1. In the case of available information, the authors should include in the column of “Properties” or “Vaccine features” for example, or including an additional column, if the approach uses delivery systems or adjuvants, identifying which ones, as well as the administration route.
- In line 84 the authors should clarify which recombinant proteins previously licensed are referring? This approach is already being adapted for large-scale production of vaccines for COVID-19?
- What does the asterisk means in the “Completion date” of the assay number EudraCT 2020-001038-36 and ACTRN12620000674932?
- It should be clarified, why are there different clinical assays under development for the same vaccine in the same country or in different countries?
- Please make clear in the Discussion section, which are the six clinical trials in phase III, indicating the vaccine type, which antigen, the use of delivery systems, adjuvants and administration route of each approach…
Author Response
Reviewer #3
Comments and Suggestions for Authors
The manuscript number 896969, entitled “Current clinical trials protocols and the global effort for immunization against SARS-CoV-2” presents a systematic review summarizing the global race for vaccine development against COVID-19 based on 50 current studies selected from the main clinical trial databases. The authors also give a global perspective of the evolution, advantages and limitations of the eight vaccine platforms against COVID-19 under clinical trial protocols based on inactivated virus; on nucleic acid as mRNA, self-amplifying RNA (saRNA) and DNA/plasmid; recombinant adenovirus serotypes platforms as adenoviral vector 5, chimpanzee adenoviral vector ChAdOx1 and combined serotypes vectors Adenovirus 5 and 26; recombinant viral protein subunits modified dendritic cells; artificial antigen-presenting cells and virus like particles. In my opinion, this manuscript is adequate for Vaccine journal since it summarizes relevant information and gives a perspective of the global efforts under development to reach as soon as possible a vaccine candidate against COVID-19, the biggest health challenge of the 21st century that affects millions of people globally. However, some points should be addressed and clarified before its publication, especially in the introduction section.
- In line 40, the authors claim there is not consolidated therapeutic drug strategy approved for COVID-19. Then, it is briefly described the use of convalescent plasma antibodies. It can also be interesting to have a critical view about some approaches suggested as potential to treat or prevent COVID-19, such as the use of chloroquine or BCG vaccine. What happens with these approaches? Could they be efficient or not?
Answer: We added more information about treatments tested in clinical trials or recommended for COVID-19 in the introduction section of this manuscript.
- Some problems of format should be improved, such as the lack of space before some references, the designation of the abbreviation CTPs should appear in line 63…
Answer: We corrected the space before references and checked all abbreviations of the manuscript.
- I feel the lack of some information to compare all the clinical trials under development. Some of this information is appearing throughout the manuscript, mainly in the discussion section, but in my point of view, the best way to summarize and compare all the approaches is to include also in Table 1. In the case of available information, the authors should include in the column of “Properties” or “Vaccine features” for example, or including an additional column, if the approach uses delivery systems or adjuvants, identifying which ones, as well as the administration route.
Answer: Thank you for your suggestion. We added more information about the delivery systems or adjuvants used in each type of vaccine in the column "vaccine features" in Table 1. The topic 3.3 was subdivided accordingly the column vaccine properties and more information about the vaccine features was analyzed associated with doses, administration route and the use of delivery systems, the age of enrolled population and their applicability in each the clinical trials. The administration route information was added in column “intervention” in Table 2, because these topics are part of a study design as well as was described in the manuscript, according to type of vaccine.
- In line 84 the authors should clarify which recombinant proteins previously licensed are referring? This approach is already being adapted for large-scale production of vaccines for COVID-19?
Answer: We have complemented the information about the recombinant protein in this mentioned paragraph.
- What does the asterisk means in the “Completion date” of the assay number EudraCT 2020-001038-36 and ACTRN12620000674932?
Answer: These clinical trials did not mention the completion date of the study, so we have not calculated their progression rate. We have added a note in the table legend explaining the meaning of the asterisk.
- It should be clarified, why are there different clinical assays under development for the same vaccine in the same country or in different countries?
Answer: This is mainly due to the characteristics of each vaccine, even being from the same platform, such as the protein subunit vaccines, they are combined with different adjuvants (Advax, MF59, matrix M, AS03, CpG1018, alum) and are under development in different countries (China, USA, Australia), as well as the Chinese dendritic cell vaccines (phase I-II), NCT04276896 /ChiCTR2000030750), in which one vaccine used DCs modified by lentiviral vector system (NHP / TYF) and the other used COVID-19 epitope gene recombinant chimeric DC vaccine.
In addition, there are different CTPs for the same vaccine platform and features, also in development in the same country, but using the different type of interventions, comparing doses, concentrations, methods of evaluation, as the case of ChAdOx1 nCoV-19 vaccines by the University of Oxford in partnership with Astra Zeneca. All these differences are highlighted in the explanation of each vaccine platform.
- Please make clear in the Discussion section, which are the six clinical trials in phase III, indicating the vaccine type, which antigen, the use of delivery systems, adjuvants and administration route of each approach…
Answer: We have added more information in the discussion section about the clinical trials in phase III; and more details about type of antigen, delivery systems, adjuvants and administration route in each vaccine platform.